# Crystal Orientation and Dislocation Slip

**Malcolm Griffiths** [1,2]

1   Department of Mechanical & Aerospace Engineering, Carleton University, Ottawa, ON K1S 5B6, Canada; malcolmgriffiths@cunet.carleton.ca or malcolm.griffiths@queensu.ca; Tel.: +1-613-585-3315

2   Department of Mechanical and Materials Engineering, Queen's University, Kingston, ON K7L3N6, Canada

**Abstract:** It is a widely held belief that dislocation slip has a direct effect on crystal orientation. Some of the confusion may be attributed to semantics when researchers are referring to related effects of dislocations on crystal orientation; either elastic bending due to constraints or the creation of geometrically necessary dislocations by climb. This communication highlights the distinction between the two and discusses why what is often imagined conflicts with what is real and possible. It is demonstrated that deformation-induced changes in the orientation of crystals are primarily limited to twinning and collections of geometrically necessary dislocations (GNDs), which in the most extreme cases are sub-grain boundaries. Alternate explanations for texture changes related to dislocation slip are provided, and they challenge the notion that grains can simply rotate because of dislocation slip through some undefined mechanism.

**Keywords:** dislocations; slip; glide; climb; twinning; texture; geometrically necessary dislocations; sub-grain boundaries

## 1. Introduction

In his seminal lecture to the Metals Society in 1938 [1], Taylor described how the orientation of a deformed material changes because of dislocation slip on a plane inclined to a tensile axis. Taylor first described the change in shape of an unconstrained single-crystal sample due to dislocation slip, as illustrated in Figure 1. He described the process as one where "we imagine the slip plane as fixed, and the orientation of the [longitudinal] axis of the specimen as changing", as shown in Figure 1a,b. He then applied the same reasoning to tensile deformation of polycrystals but assumed that each grain extended with an equal amount of strain in the tensile direction to maintain the integrity of the aggregate. To achieve this conformance, he applied a different rotation to each grain in the polycrystal needed to align each individual extension (calculated from a consideration of minimum work for the applied tensile stress) with the tensile axis as illustrated in Figure 1b,c. Taylor's hypothesis forms the basis of present-day models on crystal rotation due to dislocation slip [2].

For the polycrystal, Taylor determined the "sum of shears" along the "axis of extension" for each grain by considering the operation of five independent slip systems for the least work carried out [3]. He then calculated "the rotation of the crystal axes due to the five shears… in exactly the same way as the rotation due to single slipping" and postulated that each grain simply rotated to maintain the integrity of the polycrystalline aggregate. Taylor claimed that X-ray diffraction data showed a tendency for textures to evolve to a state where the grains re-orient to a condition where the extension axis of each grain (AC in Figure 1) becomes aligned with the tensile axis. The mechanism by which the rotation could occur in the polycrystalline case was not discussed. At no point did Taylor mention twinning as a possible mechanism for texture evolution. It may have been that twinning, which is also a shear process, was not a well-recognised mechanism for crystal re-orientation at that time. Twinning will also tend to re-orient the crystal until deformation by slip dominates. Dislocation slip is a translation and does not directly change the orientation of the crystal

so that, once a crystal has re-oriented by twinning to a point where the resolved shear stress favours slip, no further evolution of the texture may be expected. The difference between twinning and dislocation slip is that dislocation slip is a translation of one part of the crystal relative to another over what is called a slip plane and twinning is a bulk shear involving the operation of partial dislocations on many successive slip planes. Twinning changes the orientation of the crystal, but dislocation slip does not [4].

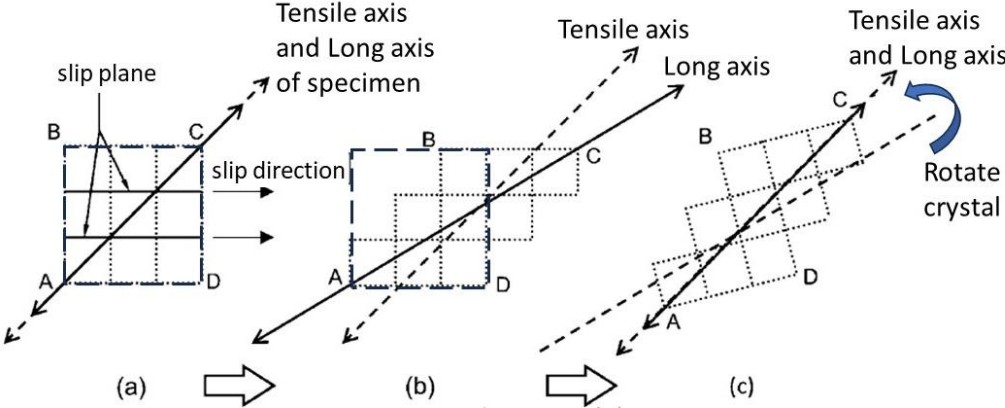

**Figure 1.** Illustration of Taylor's crystal rotation by re-defining the specimen orientation after tensile deformation of a single crystal: (**a**) undeformed crystal ABCD with the tensile axis and the longitudinal axis of the specimen (AC) illustrated by the dashed and solid lines, respectively; (**b**) translation of unit cells by slip resulting in rotation of the direction AC, which now defines the axis of extension of the crystal, but not the crystal orientation; (**c**) rotation of the crystal so that AC is aligned with the tensile axis.

It is unfortunate that Taylor's analysis has been used by many researchers as the basis for modelling texture evolution in polycrystalline samples based on dislocation slip and grain rotations [2,5–10] without due consideration of what Taylor was assuming when he made his calculations. It is also unfortunate that textbook analyses of the shear strains produced by slip introduce a rotation to maintain a symmetric strain tensor [4], which is unrelated to what Taylor postulated. Whereas the symmetry of the elastic strain tensor is a requirement to maintain equilibrium of the forces acting at a point, this modelling constraint does not apply to plastic strain because a tensor describing plastic strain is neither elastic nor does it represent the elastic stress tensor, which must be in equilibrium [11]. The confusion between the tensor analyses of elastic and plastic strain is further exacerbated by a lack of understanding of the concept of geometrically necessary dislocations (GNDs), which was first introduced in 1954 by J.F. Nye [12]. The lack of understanding is exemplified in cases in which researchers assume that misorientations observed at the nanoscale are somehow created by dislocation slip without a clear explanation of how the rotation of the crystal at the slip plane, observed without constraints, is induced by the passage of perfect dislocations over a single slip plane [13].

Whereas re-defining the specimen axis as the axis of extension for a single crystal (Figure 1) is one way to argue that the crystal re-orients by slip, the notion that individual grains in a poly-crystal simply rotate due to some undefined mechanism is difficult to accept and begs the question of whether there are other mechanisms to explain texture evolution that Taylor did not consider. Whereas one could envisage a situation in a polycrystalline specimen where individual grains are elastically distorted by different amounts, there is a limit to the elastic distortion that could be accommodated without yielding and is typically of the order of 0.1% strain, given the elastic moduli and yield stresses of most common metals. To put the elastic shear strain at the yield stress in context, it amounts to a tilt of the crystal of the order of 0.1°, which is barely detectable. Because grain boundary sliding is a mechanism that only applies at elevated temperatures > 0.4 $T_m$, where $T_m$ is the

melting temperature [14], and recrystallisation involves some point defect diffusion, any large textural change at low temperatures is better attributed to twinning.

The intent of this communication is to address how, and under what circumstances, gliding dislocations affect crystal orientations and the texture of polycrystalline aggregates. It will be limited to deformation-induced changes in crystallographic texture for common engineering alloys at low homologous temperatures ($<0.4\ T_m$) where super-plasticity from grain boundary sliding or recrystallisation is not an issue.

## 2. Dislocation Slip and Twinning

Dislocation glide or slip is a translation of one part of a crystal relative to another over a plane that is called the glide or slip plane. For perfect dislocations, the translation is a lattice vector and the crystal orientation is not altered by their passage. The most common slip plane in FCC metals is the close-packed $\{111\}$ plane and the perfect glide dislocation for this plane is $\frac{1}{2}\langle01\bar{1}\rangle(111)$. In this case, slip of the perfect dislocation involves two partials on the same $\{111\}$ plane, with Burgers' vectors of b = $\frac{1}{6}\langle\bar{1}2\bar{1}\rangle$ and $\frac{1}{6}\langle11\bar{2}\rangle$, that can be added together to give $\frac{1}{2}\langle01\bar{1}\rangle$, Figure 2a. The first of these so-called Schockley partial dislocations introduces a stacking fault on the slip plane, but this fault is then erased by the passage of the second partial that returns the crystal to perfect stacking. The crystal structure either side of the slip plane is unaltered by the passage of the Schockley pair. For the $\{111\}$ planes in FCC crystals, it is possible for partial (twin) dislocations to operate collectively on each successive plane. Each displacement on each plane is equal to a Schockley partial. Over three planes, the atom on the third layer has been translated by $\frac{1}{2}\langle11\bar{2}\rangle$, which is a vector between atoms in the perfect lattice, relative to its starting position (Figure 2a). The sheared crystal has an orientation that is a mirror image of the parent material, where the twin plane is the mirror plane and is easily envisaged for twinning on $\{111\}$ planes in face-centred cubic (FCC) metals [15,16], Figure 2b.

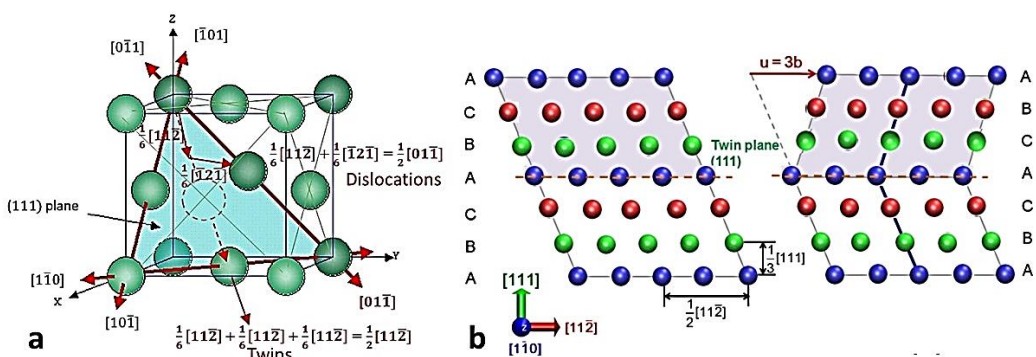

**Figure 2.** (**a**) Slip on $\{111\}$ in FCC metals showing displacement vectors for partial and perfect dislocations; (**b**) twinning by shear of **b** = $\frac{1}{6}\langle11\bar{2}\rangle$ on three successive $\{111\}$ planes results in a total displacement of **u** = 3b = $\frac{1}{2}\langle11\bar{2}\rangle$ to bring the A-layer back into register. The projection is that corresponding with the matrix $\langle1\bar{1}0\rangle_M$ in a direction coming out of the paper. The $\{111\}$ stacking layers are shown in different colours. Adapted from [16] and reproduced with permission from Elsevier.

The most common twinning planes in hexagonal-close-packed (HCP) crystals are the non-basal planes $\{10\bar{1}1\}$, $\{10\bar{1}2\}$, and $\{11\bar{2}2\}$ [17–19]. These planes are twin planes because they can each accommodate partial shear displacements (by the same displacement vector) on successive parallel planes. In so doing, the sheared planes form a perfect crystal that has a different orientation to the parent matrix. The stable positions for partial dislocations for different planes in Ti-6Al-4V were calculated by Jones and Hutchinson and are shown schematically in Figure 3 [20]. Only certain planes can accommodate twinning, i.e., those for which it is possible to shear by the same partial Burgers vector. The basal plane in HCP metals is not a twin plane because the same partial cannot operate on each consecutive plane and a given partial shear, e.g., $\frac{1}{3}[10\bar{1}0](0001)$, is only possible

on every second plane. In that case, the crystal structure changes from HCP (ABAB...) to FCC (ABCABC...). The shear operation on every second plane is thus a martensitic transformation.

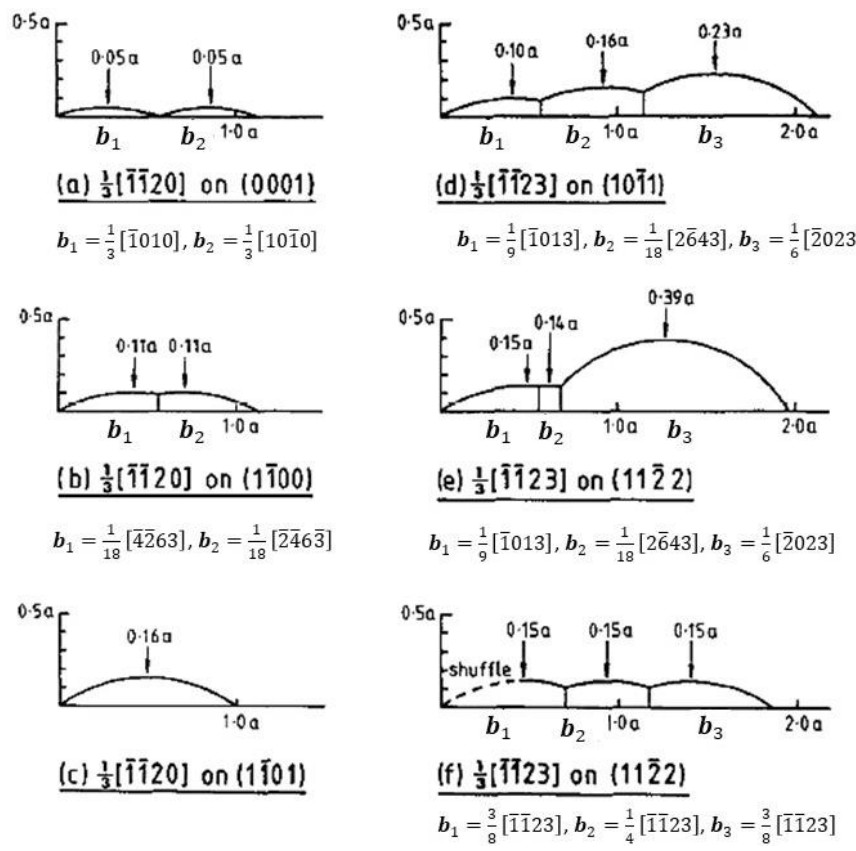

**Figure 3.** Dilatations associated with the slip process in various systems in hexagonal metals, approximated by a hard sphere model. Partial dislocations have been assumed; these cause minimum dilatation. Reproduced with permission from Elsevier [20].

For body-centred cubic (BCC) metals, the {110} plane, which is the common slip plane, has relatively stable partial slip displacements for each successive plane and can thus theoretically accommodate twinning. However, this is not a plane that is recognized as a twinning plane in BCC metals; rather, it is the {112} plane that appears to be the most common [21].

### 3. Geometrically Necessary Dislocations (GNDs)

Crystal orientation changes are sometimes attributed to dislocation glide/slip that is not concerted as it is for twinning. There is often ambiguity as to whether the orientation changes are a direct result of slip or not. Whereas the crystal orientation can be altered by an excess of dislocations of like-sign, the so-called geometrically necessary dislocations (GNDs) described by Nye [12], the change in orientation is because of an excess of dislocations of like-sign that are not on the same slip plane. During Nye's treatment, he assumed that the dislocations (of like-sign) were present in a particular arrangement and proceeded to compute the plastic bending, as opposed to elastic bending due to constraints, that would occur as a result. Considering a single crystal, Nye showed that plastic bending can be accommodated when there are more half planes (or atoms) in the top part of a crystal relative to the bottom half, i.e., the top surface is longer than the bottom surface. The easiest way to achieve this situation is for there to be non-conservative motion, i.e., climb, such as that which occurs in bending creep, as shown in Figure 4a. For cold-worked material containing a high dislocation density, climb caused by the absorption or emission of vacancies in re-

sponse to bending stress will cause dislocations of different signs to separate, as illustrated in Figure 4b, thus creating an excess of dislocations of like-sign in different regions of the crystal. This type of climbing can occur at high temperatures, when vacancy point defects are both plentiful and mobile, and during irradiation when there is a high concentration of both interstitial and vacancy point defects. Climb could also occur in irradiated materials containing prismatic dislocation loops via plastic deformation when gliding dislocations sweep up the loops, forming channels [22]. The process of incorporating the loop into the glissile dislocation is effectively a climb process and, although the tensile strain from the loops is already manifested in the material, sweeping of the loops by dislocations of one sign could result in an excess of dislocations of like-sign being accumulated in one part of the crystal, thus inducing a tilt.

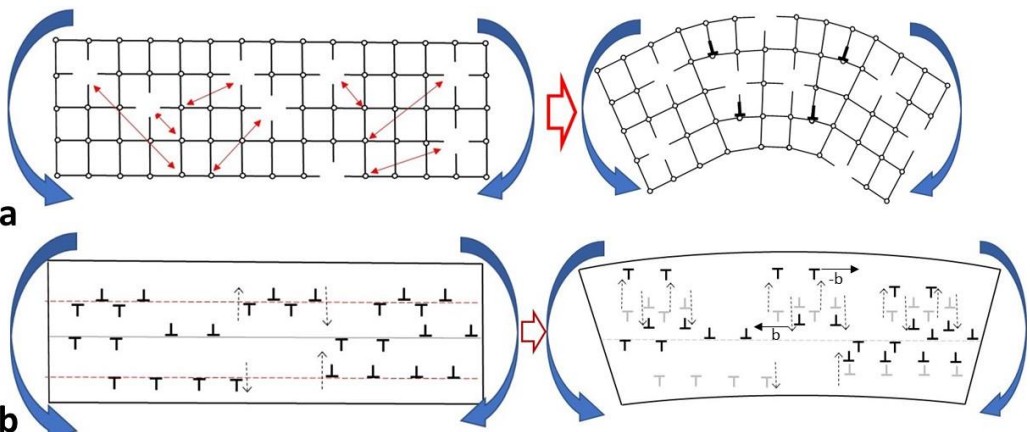

**Figure 4.** (**a**) Schematic reproduction of Nye's geometrically necessary dislocations in a bent crystal introduced by non-conservative climb in response to a bending stress (blue arrows). The progression from unbent to bent crystal (caused by climb) is denoted by the red arrow. Vacancies migrate to enable dislocation climb so that the top of the crystal is longer than the bottom by n**b**, where **b** is the Burgers' vector and n is the number of dislocations. Atoms are illustrated by small circles and atomic bonds by lines. Vacancies are denoted by the absence of an atom; (**b**) climb of dislocations with Burgers' vectors of different sign (**b** and −**b**) in a cold-worked crystal subject to a bending stress. The progression from unbent to bent crystal (caused by climb) is denoted by the red arrow.

In contrast to bending creep that is driven by dislocation climb and is thus non-conservative, deformation by slip is volume conservative, as shown in Figure 5. In Figure 5, the bottom surface is displaced relative to the top surface by the movement of the dislocations, which involves a translation, not a rotation. Although there is a shape change in the bulk material associated with the shear of numerous slip planes, there is no change in crystal orientation. The crystal can bend when there is a particular arrangement of GNDs such as that which occurs in bending creep (Figure 4). Climb of existing dislocations can occur that redistributes the atoms in the crystal so that there is an excess of half planes in the part that is undergoing tension and a deficit of half planes in the part that is undergoing compression. The excess dislocations of one sign can form sub-grain boundaries, as shown in Figure 6 [23]. Sub-grain boundaries can form to reduce the strain energy in a crystal through dislocation climb and glide. Sub-grain boundaries are arrangements of dislocations that minimize the elastic interactions between the dislocations, Figure 7a [24]. The simplest sub-grain boundary is a tilt boundary, comprised of an array of edge dislocations. The tilt of the crystal and the atomic arrangement of dislocations in a polygonised wall has been demonstrated by Zhang et al. [25] using high-resolution electron microscopy, as shown in Figure 7b. Dislocations that are free to climb and glide can take up low-energy positions relative to one another. The interaction force for a given separation of slip planes ($y$) is shown in Figure 7a for different separations on the slip plane ($x$). The dislocations have a minimum interaction force (low energy configuration) when they are arranged directly

one above the other as shown in Figure 6b but also when they are arranged in such a way that the lateral separation is equal to the vertical separation and the dislocations form a boundary that is 45° to the slip plane.

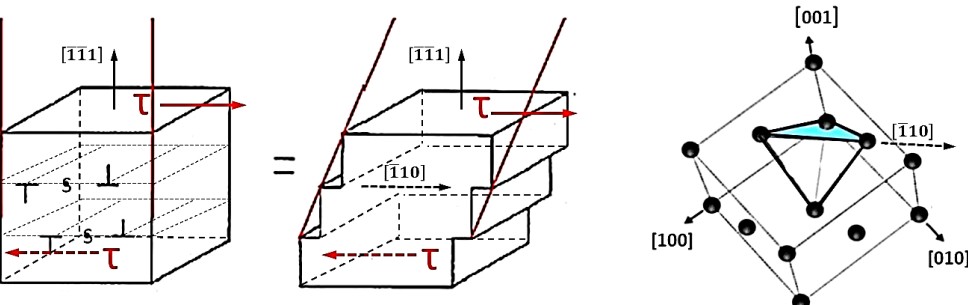

**Figure 5.** Shear of a single crystal via dislocation slip under the action of an applied shear stress. The passage of dislocations on different slip planes preserves the dimensions of the top and bottom surfaces. There is a shape change that is a series of atomic scale steps at the microscopic level but can be described as a simple shear at the macroscopic level. The shear induces the same shape change irrespective of the location of the sources (S) on the slip plane. The step width is equal to the Burgers' vector, $\mathbf{b} = \frac{1}{2}\left[\overline{1}10\right]$.

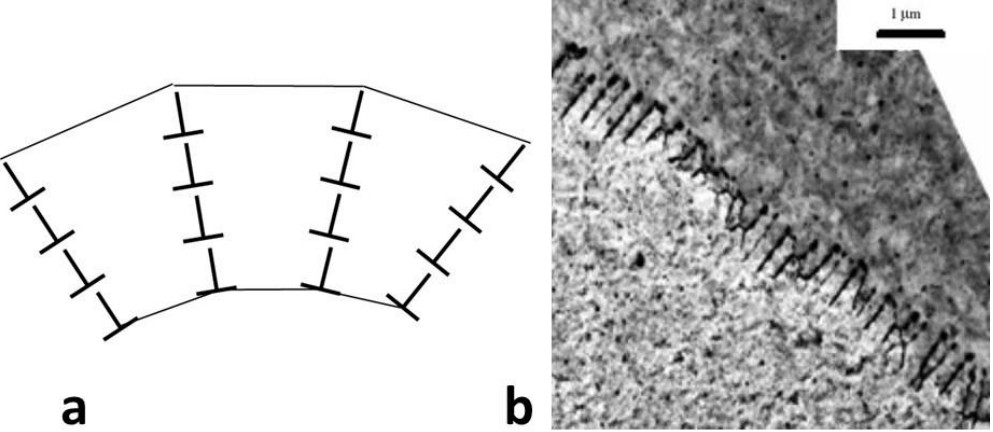

**Figure 6.** (**a**) Illustration of distribution of edge dislocations of like-sign in low-energy sub-grain tilt boundaries. (**b**) Transmission electron microscope image of a low-angle tilt boundary in a GaSb crystal. The image illustrates the effect of the tilt on diffraction contrast, which is different either side of the boundary. Reproduced with permission from Elsevier [23].

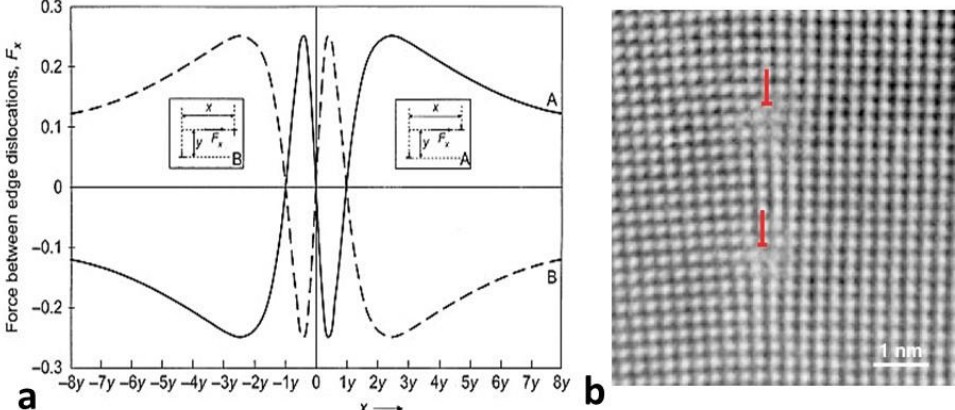

**Figure 7.** (**a**) Elastic interaction energies for dislocations as a function of relative position. The interaction energy is a minimum (zero force) when dislocations of either like-sign or unlike-sign lay

directly above one another (x = 0). The solid line corresponds with dislocation configuration A and the dashed line corresponds with dislocation configuration B. Reproduced with permission from Elsevier [24]. (**b**) High-resolution electron microscopy image showing atomic arrangement and location of dislocation half planes (in red) in a sub-grain tilt boundary in a crystal of SrTiO$_3$. The dislocations are of like-sign and are aligned vertically perpendicular to the dislocation slip plane. Reproduced with permission from APS [25].

Whereas dislocation slip alone does not change the orientation of a crystal, it is possible to create a situation where, through dislocation climb, a tilt may be induced in the crystal such as occurs in bending creep. The accumulation of dislocations of like-sign necessary for bending may also be obtained during deformation of irradiated materials via the interaction of gliding dislocations with prismatic dislocation loops creating channels.

## 4. Channelling

Although the bulk shear that occurs during twinning has a large effect on crystal orientation, there are other instances where bulk shear can occur in irradiated materials. One of the characteristics of irradiated materials containing a high density of prismatic dislocation loops is channelling [22,26–35]. Gliding dislocations sweep up prismatic dislocation loops and thus climb as they pass through a volume containing the loops. The net effect is to have bulk, as opposed to planar, shear (like twinning) but without the associated change in crystal orientation caused by the passage of successive partial twin dislocations. The passage of the gliding dislocations cannot directly re-orient the crystal even when the shear is on different slip planes (Figure 5).

Channelling could induce a rotation, however, if the climb associated with the sweeping up of prismatic dislocation loops was to create an array of dislocations of like-sign that are on different planes, i.e., somewhat similar to a sub-grain tilt boundary, Figure 8 [22,34]. The climbed dislocations of like-sign (circled by the red-dashed lines in Figure 8) are effectively GNDs that induce a tilt of the crystal. The dislocation wall (sub-grain boundary) that forms where a channel terminates, at a grain boundary for example, and any collection of dislocations of like-sign, will induce a tilt of the crystal. Such a tilt could be mis-interpreted as an elastic stress concentration [36–38] rather than the minimization of the elastic stress that would be expected from a sub-grain boundary. Irrespective of what dislocation arrangements may or may not exist in a dislocation channel, any collection of climbed dislocations, however it has formed, will affect the local orientation of the crystal.

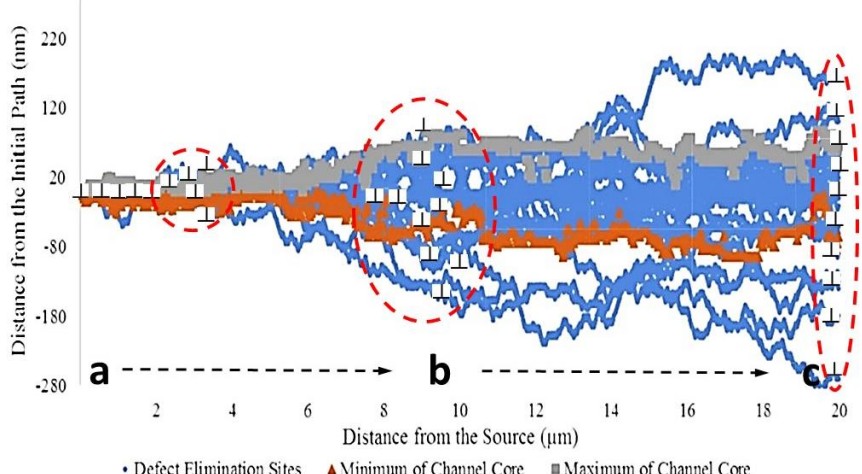

**Figure 8.** Channel developed from 100 dislocation passes with defect cluster parameters d = 10 nm, N = 5 × 10$^{23}$ m$^{-3}$ along a 20 µm path. Loop elimination is indicated in blue at each point (grey and orange demarcate the main channel): (**a**) the dislocations of a given sign are emitted from a source; (**b**) the dislocations climb (up or down) according to the vacancy/interstitial character of the prismatic

dislocation loops absorbed; (**c**) the gliding dislocations encounter a barrier and are either absorbed or form a polygonised wall (sub-grain tilt boundary). Volumes containing GNDs are indicated by red dashed lines. Reprinted with permission from [34], 2018, Elsevier.

Crystal tilting is conceivable if there is a mechanism for gathering an excess of dislocations of like-sign in a given volume of material, as shown schematically in Figure 4. Apart from the tilt that could occur at the head of a channel, crystal tilting at what is described as the edge of a channel has been reported by Jiao et al. [38]. There is often ambiguity concerning the microstructural features associated with tilting of a crystal. Jiao et al. [38] identified a region in a sample of irradiated 304 stainless steel (after deformation testing) that exhibited a 3° tilt (equivalent to about 5% elastic shear strain); see the circled area in Figure 9. The strain was associated with what is described as an "expanded channel". Assuming the authors were referring to the process of glide and climb associated with channelling [31], it is conceivable that climbing dislocations could induce a tilt of the crystal via the mechanism illustrated in Figure 4, i.e., via the accumulation of an excess of dislocations of one sign near the edges of the "expanded channel". Although the authors consider that such features are the result of channelling of the type illustrated in Figure 8 and described in numerous publications [26–35], the features marked by arrows in Figure 9 do not have the characteristics of channels (layers of material parallel to a dislocation slip plane that are cleared of dislocation loops). It is conceivable that the features marked with arrows in Figure 9 are the vestiges of the thermo-mechanical processing of the material prior to testing, i.e., possible sub-grain tilt boundaries. Either way, whether or not the tilt boundaries shown in Figure 9 were present before or after post-irradiation mechanical testing, it is likely that the strain detected is plastic in nature rather than elastic, the latter requiring the imposition of inconceivably high stress.

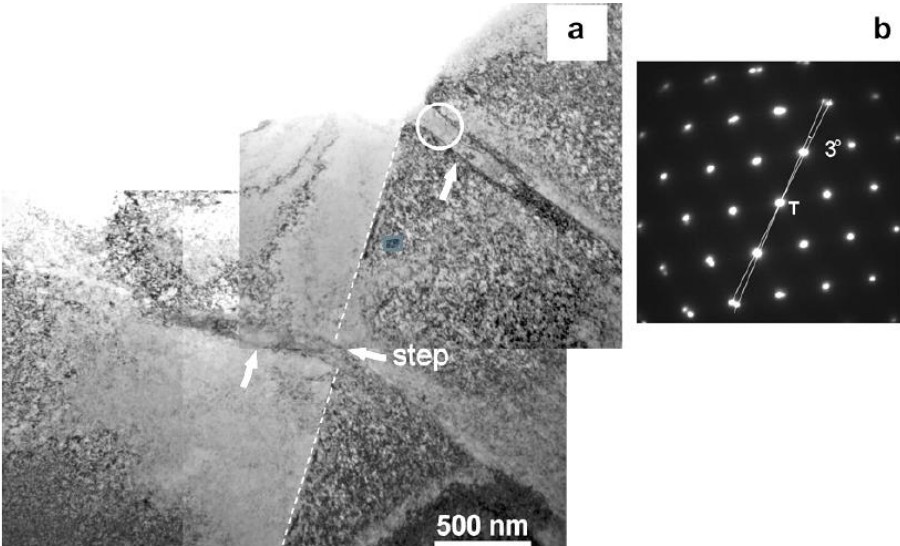

**Figure 9.** (**a**) Bright field TEM image of two expanded channels intersecting a grain boundary in alloy E irradiated to 5.5 dpa and strained to 7%, (**b**) selected area diffraction pattern from the upper expanded channel. Arrows indicate the expanded channels and a step at grain boundary. Dashed line indicates the position of the grain boundary. Reprinted with permission from [38], 2007, Elsevier.

Sub-grain boundaries can be composed of either edge dislocations (tilt boundaries) or screw dislocations (twist boundaries). There is a limit to the crystal misorientation resulting from sub-grain boundaries (typically <15°) because for larger angles, the distance between the dislocations becomes smaller than the distance where there is a large distortion from the dislocation core, typically deemed to be about 5b, where b is the length of the Burgers vector.

An association between simple slip on a single plane and crystal lattice rotation has been made in some cases for materials deformed after irradiation [13,35–38]. There is often ambiguity in the literature concerning the difference between channelling, which involves a bulk shear, and slip bands, which are planar shears [22]. Often, shear bands on a single plane are called channels and thus can be confused with the bulk clearing of radiation damage (in the form of dislocation loops) by dislocations gliding on many different planes [22]. This clearing creates a layer of softer material via the removal of the hardening species. Because channelling involves climb of the gliding dislocations as they progress through a field of dislocation loops, the shear that would otherwise occur on a single plane is spread out over many planes. The important point to note is that climb, which is a requirement for the bending of a crystal from GNDs, occurs in channelling. But the tilt can only happen in the region when there is an excess of dislocations of like-sign that are not on the same slip plane, whether at the edges or the head of a channel.

In the TEM image shown in Figure 6b, the tilt of the crystal affects the diffraction conditions either side of the row of dislocations in a sub-grain boundary. For a simple tilt boundary, the edge dislocations are aligned one above the other in a plane perpendicular to the Burgers' vector (Figure 7b). For dislocations gliding on the same plane, the TEM contrast (other than that from distortion near the core of the individual dislocations) is uniform in and around the slip band indicating that there is no tilting of the crystal [39–43]. Slip bands have the appearance of narrow channels in TEM images (hence the name), but this is the result of the fact that the electron transparent thin foil is sampling a thin slice of material that intersects the slip plane, as illustrated in Figure 10. Many examples of slip bands have been published [39–43], and there is no evidence that the crystal is tilted in those cases. Care must be taken not to confuse bend contours in the thin foil, which is an artefact of the specimen preparation, with any bending of the bulk crystal. Channelling (in irradiated materials) is common, but only in some cases, such as that reported by Jiao et al. [38], is there a crystal tilt at what is believed to be a channel. Without detailed TEM analysis, the nature of the tilting remains speculative.

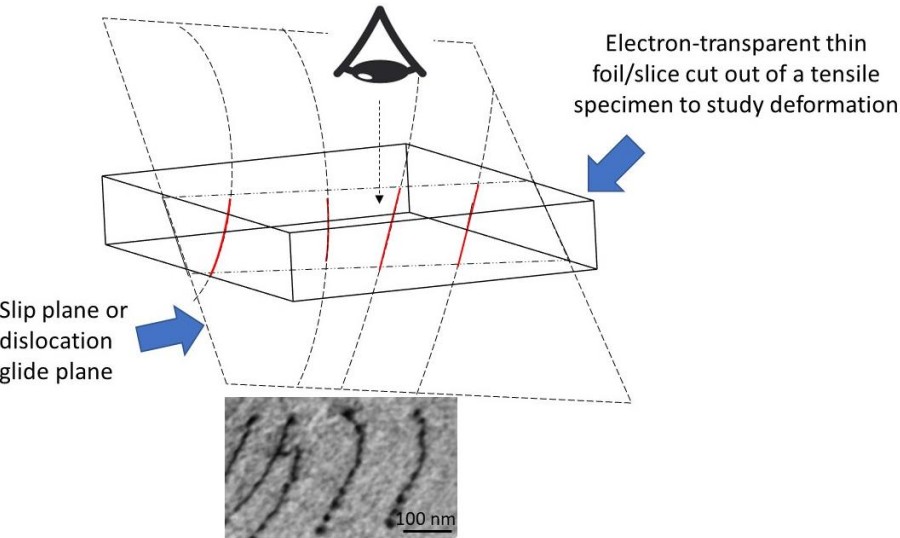

**Figure 10.** Schematic diagram illustrating how a plane of dislocations appears as a narrow band in a thin foil. The dislocations are spread over a large area but are sampled in a narrow slice. Such an image is often called a slip band. The inset shows a TEM image of dislocations in an austenitic stainless steel, reproduced with permission from Elsevier [39].

Given that channels of the type illustrated in Figures 8 and 9 involve climb of dislocations via absorption of prismatic loops, the shear from the gliding dislocations is shifted to multiple planes and the net result is a bulk, rather than planar, shear. The channels represent a bulk shear, the same as twinning, but channelling is different from twinning

because the channelling dislocations have perfect Burgers vectors and do not directly cause a change in crystal orientation, as is the case for a twin (see Figure 2). Apart from a tilt that may exist because of a so-called polygonised wall (see Figure 7), the crystal may become elastically distorted if the channel is blocked by an obstacle such as a grain boundary. Some diffuse mis-orientation will also exist if the dislocations at the end of a channel are spread over a broader volume rather than being aligned in a wall (Figure 4). The main criterion for bulk lattice distortion, i.e., bending of the crystal other than the distortion surrounding the core of a dislocation, is that there should be an excess of dislocations of one sign in the volume where the lattice is affected.

## 5. Crystal Tilting due to Constraints

A common misconception is that slip itself, which is principally a translation, induces a change in the orientation of a crystal. Whereas slip can indirectly induce elastic bending (orientation changes) of a material subject to constraints, there is no physical reason why slip, which is a translation parallel to a crystallographic plane (the slip plane), should cause a rotation of the crystal unless constraints are present. One example that is often offered to support the pre-disposition to explain bulk textural changes in terms of slip is that of tensile deformation of a crystal. In the simple case when only one slip system is activated, the orientation of the specimen must change to accommodate the dimensional change from slip, Figure 11. But the orientation change is caused by the constraints of the load frame that is rigidly attached to the ends of the specimen. Constraints that maintain the load line through the centre of the specimen and prevent rotation of the load points will induce stress that elastically bends the crystal. In some cases, such stresses may induce secondary slip systems to operate [4,22]. Without secondary slip or twinning, the specimen will elastically bend during the test. Once the constraints are removed (i.e., once the specimen is unloaded) the elastic bending stresses are relaxed and, in the absence of residual stresses, the specimen should exhibit no crystallographic orientation change with respect to the original specimen surfaces (see Figure 5). If the specimen deforms by twinning, or if the constraints are relaxed through the operation of twins, then a texture change will occur.

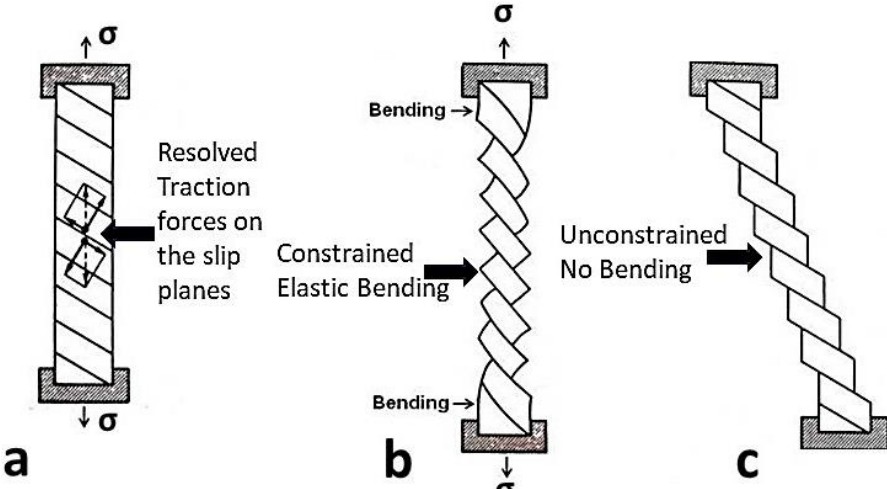

**Figure 11.** Schematic representation of tensile deformation (tensile stress—σ) of a single crystal by the operation of a single slip plane: (**a**) tensile specimen showing load points, tensile axis, and traction forces relative to slip planes; (**b**) deformation constrained by the load line inducing elastic bending strains to accommodate shear strain; (**c**) unloaded (unconstrained) specimen showing effect of elastic strain from slip. The slip planes in (**c**) are parallel with the same planes in (**a**).

Even without considering constraints, some researchers have attributed changes in crystallographic texture to dislocation slip [1,2,5–10]. This appears, in part, to be a result of the notion that the strain tensor that describes the plastic deformation must be

symmetric [2,5–10]. This is true for elastic strains for a material in equilibrium, or for infinitesimal elastic strains when not in equilibrium [11]. To maintain equilibrium of forces, the symmetry of the stress tensor (and thus the elastic strain tensor) must be symmetric. If one applies this requirement to plastic strain, this can be achieved for a system subject to simple shear (one slip system operative) by allowing the crystal to rotate. However, when using tensors to describe plastic deformation for a system that is not in equilibrium, as is the case for creep deformation, there should be no requirement for the tensor describing the plastic strain to be symmetric. The notion that a grain can rotate via grain boundary sliding in a polycrystal is also hard to conceive.

## 6. Tensor Representation of Plastic Strain from Dislocation Slip

In the book by Kelly and Knowles [4], there is an analysis of the strain in the crystal coordinate system from the shear of a plane attributed to dislocation slip. They showed how the shear strain caused by slip of a dislocation with a Burgers' vector $\mathbf{b} = (b_1, b_2, b_3)$ on a slip plane $\mathbf{n} = (n_1, n_2, n_3)$ gives a strain tensor referred to the single crystal axes that has a magnitude proportional to the amount of shear and components that are given by the product of the $\mathbf{b}$ and $\mathbf{n}$ unit vector components referred to the crystal coordinate system. What they demonstrate using vectors gives the same result as a formal tensor transformation using direction cosines, as described by Nye [11]. The details of the tensor transformation from the shear strain on a given slip plane defined by the Burgers' vector, $\mathbf{b}$, and plane normal $\mathbf{n}$ are shown in Figure 12. Kelly and Knowles show that simple shear, which is an asymmetric tensor when referred to the slip plane, could be represented by a pure shear (symmetric) tensor and a rotation, Figure 13. The resulting pure shear tensor gives a different strain tensor when transformed into the single crystal coordinate system, Figure 14. One point to note is that a symmetric tensor always remains symmetric when transformed to a different coordinate system, as illustrated in Figure 14.

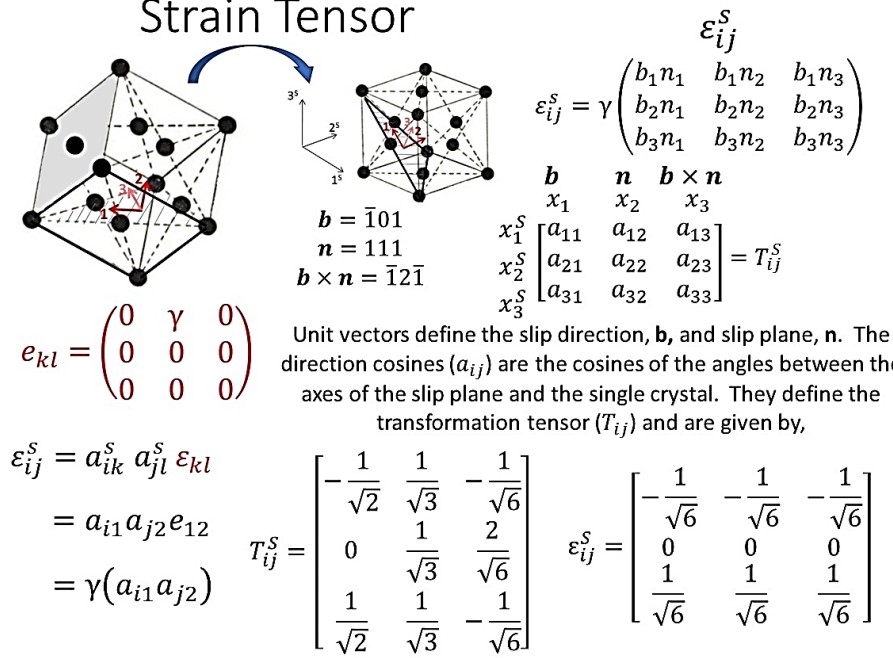

**Figure 12.** Calculation of the simple shear strain in single crystal coordinates for a given slip plane. The strain from slip on the plane defined by the coordinate system $x_i$ is transformed into the single crystal coordinate system, $x_i^S$, by the transformation tensor $T_{ij}^S$, as shown in the figure.

Consider slip on plane y (**n**) in a direction x (**b**)

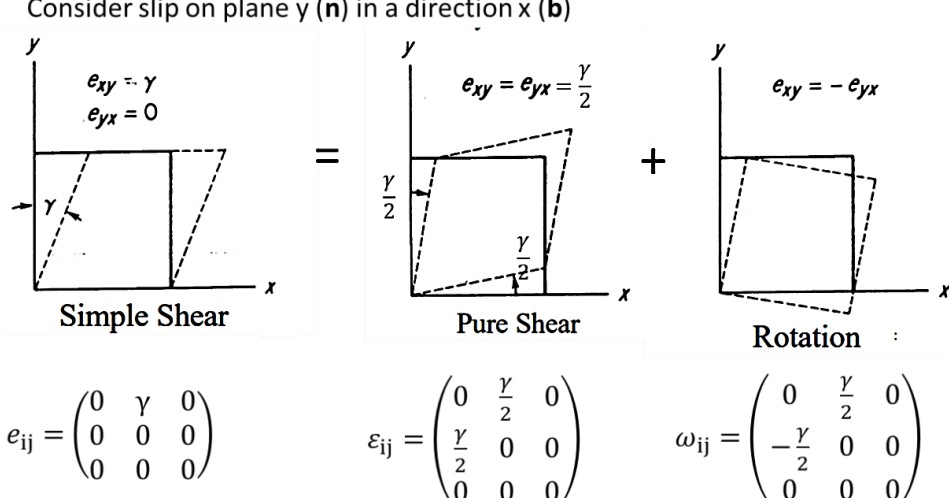

$$e_{ij} = \begin{pmatrix} 0 & \gamma & 0 \\ 0 & 0 & 0 \\ 0 & 0 & 0 \end{pmatrix} \qquad \varepsilon_{ij} = \begin{pmatrix} 0 & \frac{\gamma}{2} & 0 \\ \frac{\gamma}{2} & 0 & 0 \\ 0 & 0 & 0 \end{pmatrix} \qquad \omega_{ij} = \begin{pmatrix} 0 & \frac{\gamma}{2} & 0 \\ -\frac{\gamma}{2} & 0 & 0 \\ 0 & 0 & 0 \end{pmatrix}$$

is normal practice to separate the strain and rotation

**Figure 13.** The tensor representing simple shear on a given slip plane ($e_{ij}$) can be re-formulated as a symmetric shear tensor ($\varepsilon_{ij}$) and a rotation tensor ($\omega_{ij}$).

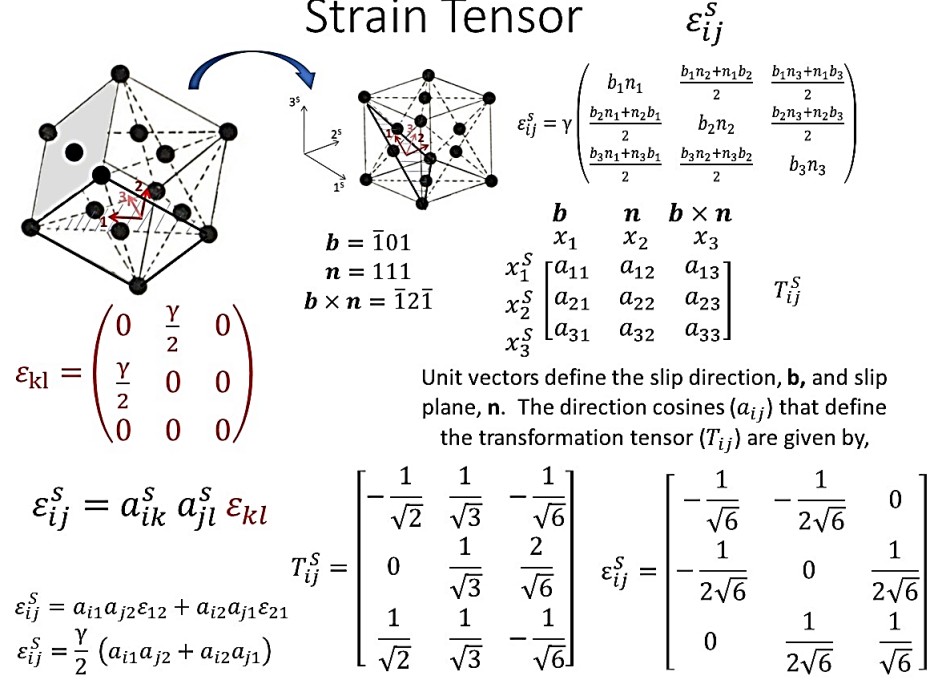

**Figure 14.** Calculation of the pure strain in single crystal coordinates for a given slip plane. The strain from slip on the plane defined by the coordinate system $x_i$ is transformed into the single crystal coordinate system, $x_i^S$, by the transformation tensor $T_{ij}^S$, as shown in the figure.

The rotation introduced to represent the simple shear tensor as a combination of pure shear and rotation tensors is a mathematical construct. Shear of one part of a crystal relative to the adjacent part by dislocation slip is a translation and does not involve crystal rotation, as illustrated schematically in Figure 5. In creating a pure shear tensor, the plastic strain tensor is treated as if it is an orthogonal system, even though the slip planes are not orthogonal (Figure 5). Even without rotation, it is not possible to generate a pure shear strain from slip in anything other than a simple cubic system. Nye [11] considered that one must convert a simple shear into a pure shear and a rotation because it is a requirement for elastic stress and strain tensors in equilibrium. However, for crystal plasticity, one does

not need to consider an equilibrium elastic state because the system being described by the tensor is neither elastic, nor is it in equilibrium.

Unfortunately, Kelly and Knowles [4] refer to a rotation produced by the glide, which gives the impression that glide causes a crystal rotation. They state that the tensor representing a simple shear is "... a pure strain tensor $\varepsilon_{ij}$ and also describes the rotation produced by the glide". This is unfortunate wording, as it implies that slip causes a crystal rotation, even for an unconstrained single crystal, which is incorrect. They describe pure shear, which is physically impossible in anything other than a pure cubic material such as common salt, by combining a simple shear with a rotation (Figure 13), They then state that "The rotation produced directly by slip does not rotate the lattice", which is correct, but they admit that in creating the symmetric tensor, they must assume that each grain may rotate freely. They then state that "whether this is so or not can only be decided by a consideration of the constraints at the inter-crystalline boundaries". Ultimately, this means that anyone working with a symmetric tensor to describe the strain due to dislocation slip must either include or disregard a rotation component when describing the evolution of the strain and crystal orientation.

The analysis described by Kelly and Knowles [4] may still lead some to the mistaken impression that slip alone changes the crystal orientation. As they note, glide (slip) is a translation, and any rotation of the crystal axes that could occur has to come about because of constraints imposed on the crystal, such as those that might be experienced in a tensile test (see Figure 11). In that simple case, the constraints imposed by the load frame induce bending stresses on the specimen, thus changing the crystal orientation elastically. It is only the constraints from the load frame that bend the tensile specimen elastically and the bending disappears when the constraints are removed, i.e., there is no plastic rotation due to slip of the unconstrained crystal.

## 7. Tensor Representation of Orientation Change with Twinning

It could be argued that twinning, which involves the passage of partial (twin) dislocations on successive planes through the material, is a dislocation slip process. It is, but it is a special shear process, because it is a bulk shear and the sheared crystal has a specific orientation relative to the parent matrix. The transformation from crystal to twin is given by the tensor operations relating the crystal axis system to the twin axis system in Figure 15. For a {111} twin in an FCC crystal, the transformations from crystal to twin are achieved by transforming the original crystal matrix axes $[001]^M$, $[010]^M$ and $[100]^M$ to the axes defining the twin plane and shear direction in the crystal, $[1,\bar{1}.0]$, $[1,1,\bar{2}]$ and $[1,1,1]$. The new axis system is then transformed into axes defining the twin interface; this involves rotation about the $[1,1,1]$ axis by $180°$. From there, the axis system defining the twin interface is transformed to the axes defining the FCC crystal in the twin, i.e., $[001]^T$, $[010]^T$, and $[100]^T$. The transformation tensor relating vectors in the matrix to the twin is shown. For any plane or direction vector in the matrix, the index of that vector in the twin is given by the transformation shown. The orientation relationships of base cubic axes in the twin and matrix are illustrated in Figure 15. One can see that some low index orientations in the matrix have similar low index orientations in the twin, and there are some low-index zone axes, e.g., $[111]^M\|[111]^T$, $[1\bar{1}0]^M\|[\bar{1}10]^T$ and $[11\bar{2}]^M\|[\bar{1}\bar{1}2]^T$, plus others where the twin and matrix are parallel. The relative orientations have been illustrated by Bleikamp et al. [15], and modified versions of their illustrations are shown in Figure 16. This figure illustrates that the twin crystal structure is a mirror image of the matrix for the (111) and $(1,1,\bar{2})$ planes.

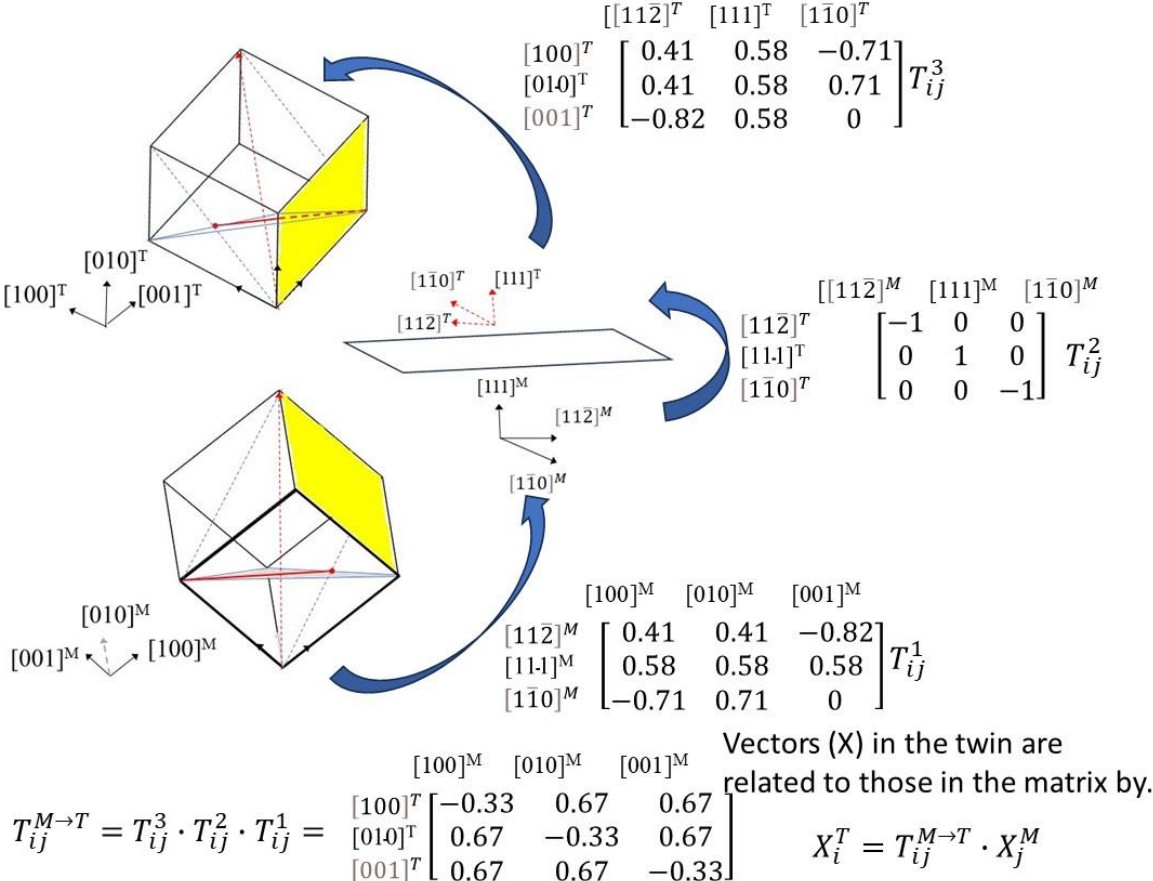

$$T_{ij}^{M \to T} = T_{ij}^3 \cdot T_{ij}^2 \cdot T_{ij}^1 =$$

Vectors (X) in the twin are related to those in the matrix by.

$$X_i^T = T_{ij}^{M \to T} \cdot X_j^M$$

**Figure 15.** Direction cosines for the transformation matrix for vector in the base crystal matrix (M) to a twin (T) for the {111} twinning plane in FCC metals. The matrix-to-twin transformation is easiest to determine by transforming to an intermediate axis system parallel to the twin interface related by the direction cosines shown.

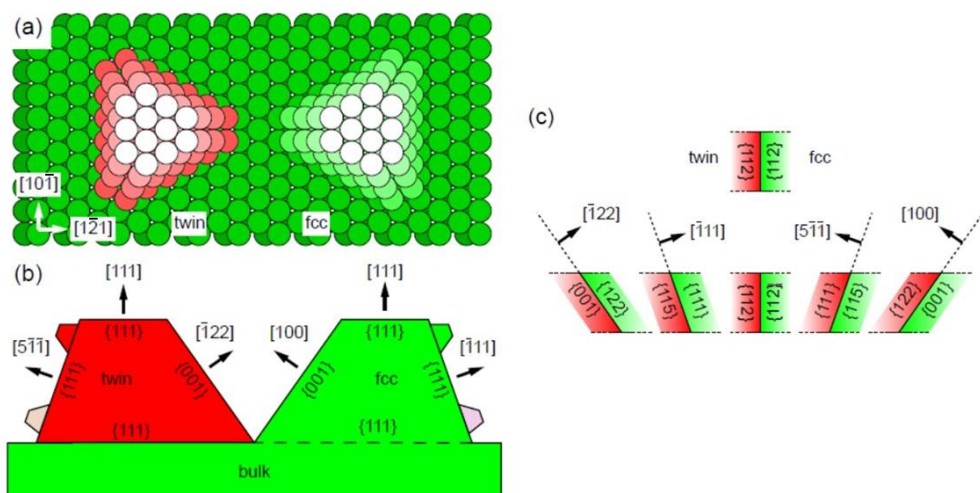

**Figure 16.** Orientation relationship between matrix and twinned volume for {111} twinning in FCC metals: (**a**) ball model looking down [111] showing stacking in twin (red) and matrix (green) faceted by {001} and {111} planes; (**b**) directions relative to the matrix for the facets shown in (**a**); (**c**) orientation relationship between planes in crystal matrix and twin. Modified from [15] with permission from APS.

## 8. Crystallographic Texture and Deformation Twinning

By far the most common, and likely, cause of a change in orientation of a crystal is twinning. Because cubic materials are relatively isotropic in their properties, there has been little work on the effect of deformation on the textures of polycrystalline cubic materials. There have been, however, numerous studies on texture evolution due to twinning in HCP metals [8,44–52], and more recently for austenitic alloys [53–56]. There have been a number of modelling studies that attribute texture evolution to dislocation slip, where researchers incorporate a rotation by imposing a requirement that the plastic strain tensors must be symmetric [2,5–10]. Experimental observations show that the primary cause of texture changes in HCP metals is twinning, although some researchers have developed models for texture evolution based on slip as well as twinning [8]. The imposition of an elastic strain tensor, which is always symmetric, is a requirement for equilibrium of forces but is not necessary when considering plastic strain, which by its nature is not an equilibrium process. The notion that grains can physically rotate is hard to accept given that most engineering alloys such as Zr-2.5Nb pressure tubing have complex platelet-like grain structures, Figure 17. Although there are many sub-grain boundaries from thermo-mechanical processing in these components, those boundaries are mostly present to accommodate small misfits between the transforming beta-phase and matrix during cooling of this dual phase alloy. The notion that grains might rotate by a slip-based mechanism is difficult to comprehend unless it involves GNDs. Most textural changes during fabrication of Zr-alloys can be attributed to twinning [44–52]. An example of texture changes caused by the operation of different twinning systems in the simple case of rolling is shown in Figure 18. Samples of plate Zr-2.5Nb with a microstructure like Zr-2.5Nb pressure tubing shown in Figure 17 were cold-rolled in the longitudinal and transverse directions. For the transverse rolling, there was a change in the basal (0002) texture coefficients that can be explained primarily by the operation of $\{\bar{1}012\}\langle\bar{1}01\bar{1}\rangle$ and $\{11\bar{2}1\}\langle\bar{1}\bar{1}26\rangle$ twinning systems [17,47,48,51]. The effect of twinning is to produce a decrease in basal pole intensity at one angular orientation while simultaneously increasing the basal pole intensity at a separate location with a characteristic rotation of the lattice dependent on the twin plane and direction. The thin platelet-like grain structure in Zr-2.5Nb pressure tubing makes it difficult to clearly distinguish and image twinning, but the texture changes can be attributed to twinning based on the characteristic stepwise change in texture intensities at particular orientations. If texture changes were the result of a lattice rotation caused by dislocation slip, one would expect a broad spread of orientation changes, which is not apparent (Figure 18). Using the Kearns' basal texture parameters [17], which give a measure of the tendency for basal poles to be oriented towards the radial ($f_R$) and transverse directions ($f_T$), Figure 18b, one can see that the basal texture changes for rolling in the transverse, but not the longitudinal, direction. The lack of a basal texture change when rolling in the longitudinal direction is because there were very few grains oriented with their c-axes close to the main strain axes, i.e., the plate normal and rolling directions. Unfortunately, the prism pole texture was not measured and one cannot then comment on the possibility of prism texture changes from either twinning or slip for this orientation. There was, however, some additional work examining the dislocation structures for the two orientations from tensile deformation of the same plate material. Both twinning and pyramidal c + a slip was observed for the tensile axis parallel with c, but only prism a-slip was observed when the tensile axis was perpendicular to the c-axis [52].

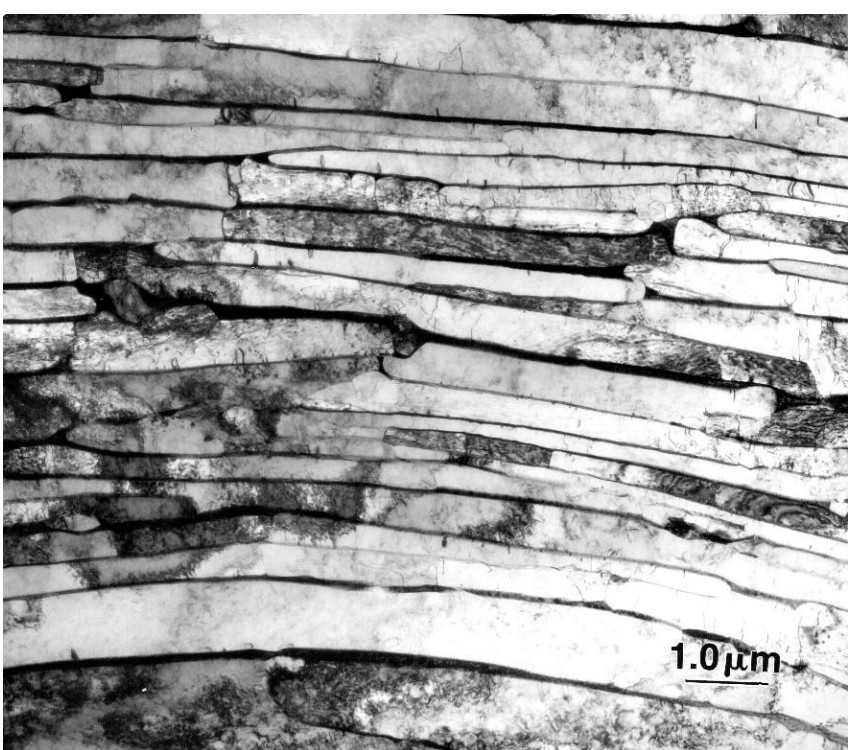

**Figure 17.** TEM image of grain structure in Zr-2.5Nb pressure tubing looking along the axis of the tube. The grains are thin platelets and not conducive to rotation. Unpublished data reproduced with permission from the CANDU Owners Group (COG) and CNL.

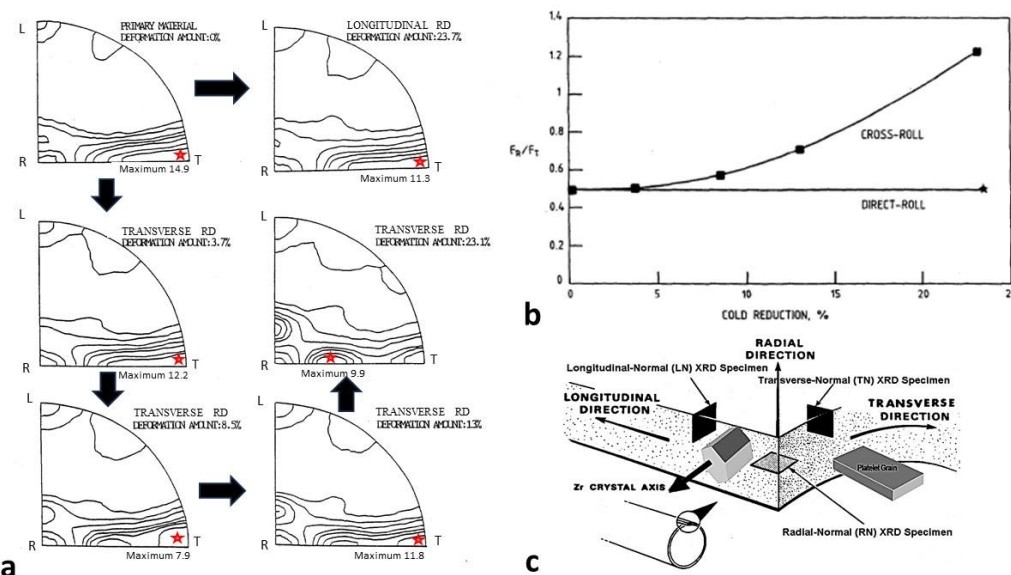

**Figure 18.** Texture evolution of Zr-2.5 wt% Nb plate material following rolling in the longitudinal and transverse rolling directions (RD): (**a**) quarter pole figures for the basal pole (c-axis) pole as a function of rolling strain. The contours depict lines of equal basal pole diffraction intensity relative to a random sample with maxima marked by the red stars. The radial, transverse, and longitudinal specimen directions are denoted by R, T, and L, respectively; (**b**) ratio of the radial ($f_R$) and transverse ($f_T$) Kearns' texture parameters as a function of strain for the radial and transverse RDs; (**c**) schematic showing main texture component and orientation of specimens for X-ray diffraction (XRD). Unpublished data reproduced with permission from the CANDU Owners Group (COG) and CNL.

Ultimately, there is no definitive physical evidence for a slip-based crystal re-orientation. The possibility of such an occurrence physically occurring is rather weak, given that most deformation processes being considered are for engineering alloys that have complex grain structures not conducive to rotation of individual grains.

The dichotomy between slip and twinning as deformation mechanisms has been raised in connection with texture changes observed during compression of annealed Zr-2.5Nb material. Salinas-Rodriguez [8] studied the evolution of texture in Zr-2.5Nb rod material during compression testing. He reported on the re-orientation of grains that had their c-axes perpendicular to the compression axis for two different grain sizes, 5 μm and 10 μm in diameter. Both exhibited twinning, but the larger-grained material exhibited significantly more twinning than the smaller-grained material and the texture change was larger as a result for approximately the same strain (about 15–20%). Salinas-Rodriguez attributed the basal pole texture change for both materials to $\{\bar{1}012\}\langle\bar{1}01\bar{1}\rangle$ twinning and the slower rate of texture change (as a function of strain) in the finer-grained material to a higher preponderance of $\frac{1}{3}\langle1\bar{2}13\rangle\{\bar{1}101\}$ c + a dislocation slip in that material, i.e., more slip and less twinning. The reason for the preponderance of c + a slip in the smaller-grained material was attributed to higher oxygen content in the coarser-grained material that inhibited c + a dislocation motion, thus resulting in more twinning.

Salinas-Rodriguez attributed a small change in prism pole texture to dislocation slip, although that deduction was confounded by noting that the intensity of the prism plane diffraction peaks may also be affected by twinning that can affect the prism pole as well as basal pole texture. He reported that the prism pole texture representing most of the grains rotated 30 degrees after 20% cold-working for the smaller-grained material only. No such prism texture change was observed for the larger-grained material that was deformed up to 15%. Although Salinas-Rodriguez attributed this marked shift in prism texture to $\frac{1}{3}\langle1\bar{2}10\rangle\{\bar{1}010\}$ dislocation slip, the texture shift (by 30 degrees) is one that is commonly observed during recrystallisation [17]. Any stepwise change in prism pole $(\bar{1}010)$ texture is an indication of either recrystallisation or twinning rather than slip. The data plotted in [8] for the finer-grained material is confounded by the sharp prism pole texture change (a 30-degree rotation) for 20% strain that then disappeared at higher strains. The reason for this is not known and has not been explained [8].

The evolution of the prism pole texture was better behaved for the coarser-grained material deformed up to 15% strain, as shown in Figure 19. The important point to note is that (prior to deformation) there is a peak in $(\bar{1}010)$ texture intensity both parallel and at 60 degrees to the compression axis, consistent with a radial distribution of basal poles peaked at about 90 degrees to the compression axis. The $(\bar{1}010)$ texture component parallel with the compression axis decreases with increasing strain but is simultaneously increasing for angles that are approximately 90 degrees to the compression axis. The prism pole intensity increase at 90 degrees to the compression axis is consistent with $\{10\bar{1}2\}\langle\bar{1}011\rangle$ twinning for grains that are oriented with basal poles at 90 degrees to the compression axis, Figure 19. Note that (from symmetry) there will always be $(\bar{2}020)$ diffraction peaks in intensity at 60 degrees from each other for the same grain, but there is only one unique $(0002)$ diffraction peak at a particular orientation for any given grain. In this case, the changes in $(\bar{1}010)$ texture intensity are consistent with the effect of $\{10\bar{1}2\}\langle\bar{1}011\rangle$ twinning rather than dislocation slip.

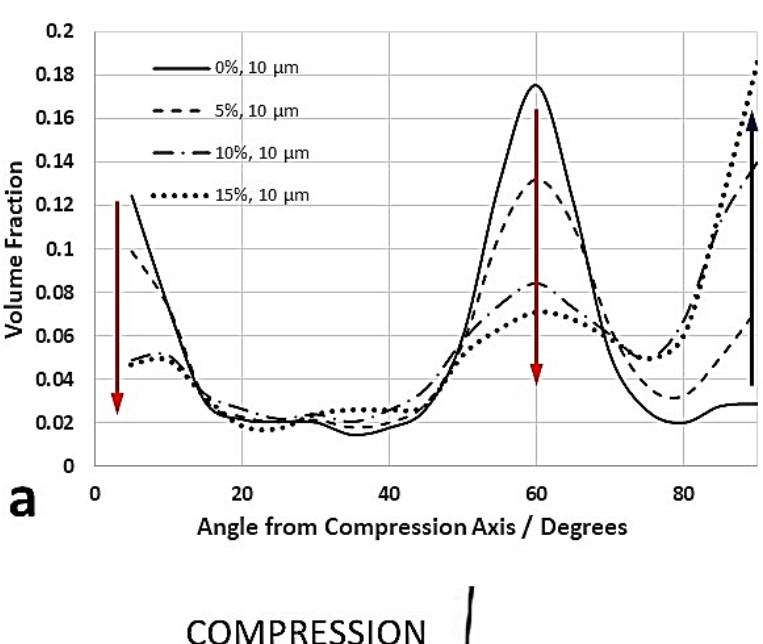

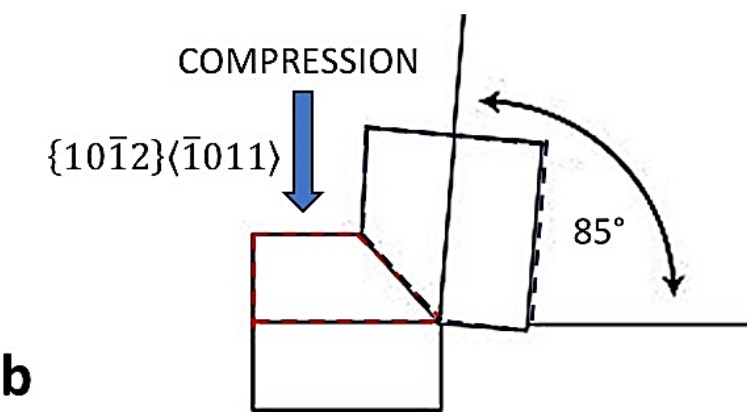

**Figure 19.** (**a**) Plot of prism pole [$\bar{1}010$] texture as a function of % strain in compression. The prism pole texture peaks that correspond with the compression axis and thus at 60 degrees from this axis due to symmetry decrease with increasing strain (red arrows), with a corresponding increase in texture intensity at 90 degrees from the compression axis (blue arrow). Modified from [8] and reproduced with permission from Elsevier. (**b**) Schematic showing the twinning re-orientation for grains that have [$\bar{1}010$] poles parallel with the compression axis. When twinned, the prism pole texture increases at 90 degrees to the compression axis (see plot).

## 9. Discussion

The notion that crystal orientations rotate by slip as postulated by Taylor in 1938 [1], is a widely held belief even today [57]. In their recent article, Zepeda-Ruiz et al. [57] describe the elastic rotation of a single crystal caused by the constraints of the load frame during tensile testing, as illustrated in Figures 1 and 11. They then conclude that "On the basis of such purely geometrical considerations, Schmid predicted that under tension a crystal should rotate to align its dominant slip direction with the straining axis. Conversely, in a frame tied to the crystal lattice, the straining axis rotates towards the dominant slip direction". Although their article is directed at improving polycrystalline modelling, there is a disconnect between the rotation that is described by the elastic constraints for a single crystal and what happens in a polycrystalline material. In the same way that Taylor imagined that the individual grains in a polycrystal can rotate in a tensile test to accommodate the constraints of the loading axis, Zepeda-Ruiz et al. make the same assumptions and describe the texture evolution assuming that individual grains are free to rotate.

The elastic rotations in a load frame are commonly confused with the observation that the long axis of a single crystal deforming due to dislocation slip tends to be closely

aligned with the operative slip direction. At some point, researchers lose sight of the fact that the rotation with respect to the original crystal coordinate system in the load frame is an elastic distortion caused by constraints, while the crystal orientation with respect to the original crystal coordinate system remains the same. In terms of a polycrystalline system, the same may be said of the specimen coordinate system. If one does not change the reference coordinate system from that fixed by the original specimen axes, any rotation of individual grains must come about via grain boundary rotation.

Crystal rotation can be induced by extremely large deformations such as those that occur as a result of impacts [58], but are mostly restricted to grain boundary sliding during high-temperature creep [14] or creep of nano-crystalline materials [59]. Kelly and Knowles [4] noted that the rotation that is invoked in the tensor formulation of slip has to come about via some undefined external process (see Section 6). But it appears that many modellers simply use the mathematical construct described by Kelly and Knowles, which introduces a rotation without considering how the rotation can be applied, while quoting Taylor [1], who only imagined that crystal rotation happens in a polycrystalline material without reference to the possibility of twinning to explain texture changes. More recent solid mechanics modelling invokes twinning, which is a real rather than imaginary phenomenon, to account for texture changes during plastic deformation [60]. That being said, there are circumstances in which dislocations are responsible for changes in crystal orientation but, as explained in Section 3, the dislocations in question are known as GNDs. They account for strain gradients within individual crystals/grains of a polycrystalline material [61] and should therefore not be confounded with the large-scale changes in texture that are typically associated with twinning [15–19].

## 10. Conclusions

The orientation of an unconstrained single crystal cannot change as a result of dislocation slip alone.

A change in orientation of a crystal can occur via shear deformation in the special case of twinning that involves the passage of partial (twin) dislocations on each successive glide plane.

The crystal orientation can change when there is an excess of dislocations of like-sign. When geometrically necessary dislocations are concentrated in a given volume of material, it is necessary that the crystal exhibits some plastic bending. When aligned in low-energy configurations, the collection of GNDs is known as a sub-grain boundary.

Sub-grain boundaries can be composed of edge or screw dislocations, or both. For a boundary comprised of edge dislocations only (tilt boundary), the misorientation axis is parallel to the deformation line and within the boundary plane. For a boundary comprised of screw dislocations only (twist boundary), the misorientation axis is normal to the boundary.

Crystalline misorientations may exist in polycrystalline samples due to variations in internal elastic stresses. Elastic bending of the crystal due to constraints is limited to small strains (of the order of 0.1%) by the fact that the crystal will always yield on secondary slip systems to relieve the constraint.

Crystallographic texture changes caused by grain rotation are a theoretical construct for most cases of deformation involving engineering alloys. Crystallographic texture changes only result from twinning or when there is a collection of GNDs, such as that which exists in sub-grain boundaries. GNDs are created by non-conservative motion of dislocations (climb).

**Funding:** This research received no external funding.

**Data Availability Statement:** Data is contained within the article.

**Acknowledgments:** The author would like to acknowledge Chalk River Laboratories (now Chalk River Nuclear Laboratories (CNL)) for permission to use unpublished micrographs in Figure 17. The author is also grateful to the CANDU Owners Group (COG) for giving permission to use the data shown in Figure 18.

**Conflicts of Interest:** The author declares no conflict of interest.

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
