# Peer review of "Crystal Orientation and Dislocation Slip"

_metals, doi:10.3390/met13121950_

Round 1

Reviewer 1 Report (Previous Reviewer 2)

Comments and Suggestions for Authors

Very interesting article. I recommend it for publication.

There are, of course, controversial points regarding the explanation of deformation texture using the twinning process. The author of the work does not take into account the stacking fault energy, which contributes to the inclusion of additional slip planes and the splitting of complete dislocations. Titanium alloys are not a good example because their stacking fault energy is low. A more complex option for the proof presented by the author can be nickel alloys with an fcc structure and high packing energy, in which twinning is difficult and a deformation texture is formed. As for bcc alloys, I agree with the author that it is impossible to explain twinning using dislocation glide.

Overall, the work is very interesting and can be useful for both engineers and university teachers.

Comments on the Quality of English Language

 Minor editing of English language required. Please check the entire text, there are missing and dangling sentences.

Author Response

Reviewer #1

Very interesting article. I recommend it for publication.

There are, of course, controversial points regarding the explanation of deformation texture using the twinning process. The author of the work does not take into account the stacking fault energy, which contributes to the inclusion of additional slip planes and the splitting of complete dislocations. Titanium alloys are not a good example because their stacking fault energy is low. A more complex option for the proof presented by the author can be nickel alloys with an fcc structure and high packing energy, in which twinning is difficult and a deformation texture is formed. As for bcc alloys, I agree with the author that it is impossible to explain twinning using dislocation glide.

Overall, the work is very interesting and can be useful for both engineers and university teachers.

Thank you for the comments.  Regarding twinning, I describe twinning as it applies to the three main crystal systems – FCC, HCP and BCC.  The intent is not to discuss why a particular crystal twins and on what planes, rather to briefly note that twinning occurs for each system and that it is based on the shear of partial dislocations on each successive plane.  One of the peculiarities of the HCP system is that twinning does not occur on basal planes.  Twinning does occur on non-basal planes and I chose to reference work by Jones and Hutchinson to show what partial dislocations are possible on non-basal planes in HCP metals – the fact that there paper was on a Ti-alloy is secondary to the fact that it described the possible partial dislocations in HCPs quite well.  It is true that a stacking fault is generated at the twin interface with the main crystal but because of partial shearing occurs on each successive plane the crystal is perfect, and there are no stacking-faults, within the twin.

Reviewer 2 Report (New Reviewer)

Comments and Suggestions for Authors

The manuscript by M.Griffiths represents kind of review or discussion paper. It does not contain new experimental results, but presents rather detailed and thorough discussion of some problems in the field of materials deformation with emphasis on metals and alloys. It does deserve to be published, since at present the papers discussing basic and fairly fundamental issues are rarely encountered.

However, the presentation of the text and of figures is just terrible.The figures are messed up with the text. However, most figures are far from being acceptable. For example, the fig. 8 is barely comprehensible. Since the text of the manuscript requires frequent look at the figures, their poor quality is very annoying.

Most importantly, the conclusions, albeit being reasonable, are not directly related to the manusript text. The text discusses many different issues and it is unclear how the conclusions of the manuscript were reached.

In summary, the manuscript is scientifically sound, but in present form it cannot be published. It must be seriously reworked.

Author Response

Reviewer #2

The manuscript by M.Griffiths represents kind of review or discussion paper. It does not contain new experimental results, but presents rather detailed and thorough discussion of some problems in the field of materials deformation with emphasis on metals and alloys. It does deserve to be published, since at present the papers discussing basic and fairly fundamental issues are rarely encountered.

However, the presentation of the text and of figures is just terrible.The figures are messed up with the text. However, most figures are far from being acceptable. For example, the fig. 8 is barely comprehensible. Since the text of the manuscript requires frequent look at the figures, their poor quality is very annoying.

Apologies for the inconvenience.  Figure 8 and the associated text have been modified to be more clear.  I have also modified Figure 1 and associated text to improve the quality and clarity.  Figure 19 has also been improved/updated.

Most importantly, the conclusions, albeit being reasonable, are not directly related to the manusript text. The text discusses many different issues and it is unclear how the conclusions of the manuscript were reached.

An abstract has been included that should now make it clearer how the manuscript and conclusions are related.

In summary, the manuscript is scientifically sound, but in present form it cannot be published. It must be seriously reworked.

Thank you – have addressed as suggested.

Reviewer 3 Report (New Reviewer)

Comments and Suggestions for Authors

1.  Authors should replenish some references within the last 5 years. All the references are beyond 10 years in the article.

2. In page 3, line 10,"Recent work by " ; the sentence isn't finished, it looks like missing something.

3. In figure 19 (a) , it should be shown the units of abscissa and ordinate.

Author Response

Reviewer #3

  1. Authors should replenish some references within the last 5 years. All the references are beyond 10 years in the article.

While many of the references are historic there are some more recent references within the past 5-10 years – e.g. 10, 13, 16, 19, 30, 34. 40.  The topic is not restricted to state-of-the-art research and the ones quoted are appropriate.

  1. In page 3, line 10,"Recent work by " ; the sentence isn't finished, it looks like missing something.

Thank you – have deleted.

  1. In figure 19 (a) , it should be shown the units of abscissa and ordinate

Have added (degrees) to abscissa. Ordinate is dimensionless.

Reviewer 4 Report (New Reviewer)

Comments and Suggestions for Authors

metals-2699671
Title: Crystal Orientation and Dislocation Slip
Author: Malcolm Griffiths

This work is a sum-up of the existing literature on how gliding dislocations affect crystal orientations and the texture of polycrystalline aggregates. The article is helpful for materials science scientists.

Page 1. This paper cannot be graded as communication. The manuscript does not present the author's original research results. It is review paper that summarizing the state of research on a topic. Communication is shorter version of "Original Paper", whose methods, findings, etc. don't justify a full length paper.

In terms of scientific content there is nothing to add, considering that it is a review and not the presentation of a work conducted by the author.

Abstract and keywords are obligatory.

Ensure that all acronyms and abbreviations are defined when first used in the text.

Figure 6 was not reproduced from [22].

Sections must be numbered.

Page 12. Paper [3] was not written by Kelly and Knowles.
Page 14: Paper [1] was not written by Bleikamp et al.
It is suggested to review the accuracy of citations throughout the manuscript.

Data Availability Statement should be added.

Author Response

Reviewer #4

This work is a sum-up of the existing literature on how gliding dislocations affect crystal orientations and the texture of polycrystalline aggregates. The article is helpful for materials science scientists.

Page 1. This paper cannot be graded as communication. The manuscript does not present the author's original research results. It is review paper that summarizing the state of research on a topic. Communication is shorter version of "Original Paper", whose methods, findings, etc. don't justify a full length paper.

According to Wikipedia “Scientific communication is the practice of informing, educating, raising awareness of science-related topics, and increasing the sense of wonder about scientific discoveries and arguments.”  I chose communication as opposed to a “letter to the editors” because I want to communicate some ideas and observations concerning widely held beliefs in materials science that needed to be questioned.  I will ask the editors if there is a better category for what I have submitted.  

In terms of scientific content there is nothing to add, considering that it is a review and not the presentation of a work conducted by the author.

See response above.  While referring to other work is necessary to make certain points about which I want materials scientists the intent is not to be a review, rather a commentary for education puroposes.

Abstract and keywords are obligatory.

See previous comments.  As with a “letter to the editors” I believe a “communication” is not a conventional paper.  However, I have drafted a short abstract (with keywords) to be used by the editors if necessary as follows –

It is a widely held belief that dislocation slip has a direct effect on crystal orientation.  Some of the confusion may be attributed to semantics when researchers are referring to related effects of dislocations on crystal orientation; either elastic bending due to constraints or the creation of geometrically necessary dislocations by climb.  This communication highlights the distinction between the two and discusses why what is often imagined conflicts with what is real and possible.  It is demonstrated that deformation-induced changes in the orientation of crystals are primarily limited to twinning and collections of geometrically necessary dislocations (GNDs), which in the most extreme cases are sub-grain boundaries.  Alternate explanations for texture changes related to dislocation slip are provided that challenge the notion that grains can simply rotate because of dislocation slip by some undefined mechanism.

Keywords:  dislocations, slip, glide, climb, twinning, texture, geometrically necessary dislocations, sub-grain boundaries.

Ensure that all acronyms and abbreviations are defined when first used in the text.

I believe that is the case but if any have been missed they will be picked up by the editors in the copy-edit process.

Figure 6 was not reproduced from [22].

Apologies – a reference error – changed to [23].  Thank you.

Sections must be numbered.

Sections have now been numbered.  Thank you.

Page 12. Paper [3] was not written by Kelly and Knowles.

Apologies – changed to [4].

Page 14: Paper [1] was not written by Bleikamp et al.

Apologies – changed to [15].

It is suggested to review the accuracy of citations throughout the manuscript.

Yes, thank you for picking up the mistakes – all references have now have been checked and corrected where necessary.

Data Availability Statement should be added.

Will be added as part of the editing process.

This manuscript is a resubmission of an earlier submission. The following is a list of the peer review reports and author responses from that submission.

Round 1

Reviewer 1 Report

Comments and Suggestions for Authors

Dear Editor,

I cannot see any changes made by the author after the review. The paper needs to be extensively rewritten and restructured. There is then no reason to consider the paper for the publication in Metals.

Reviewer 2 Report

Comments and Suggestions for Authors

I recommend that the author choose one type of crystal lattice to explain his theory. 

Comments on the Quality of English Language

English needs to be checked, there are many grammatical errors.

Reviewer 3 Report

Comments and Suggestions for Authors

Minor revision is needed.

(1) the author and affiliation are missing;

(2) Figure 1, the image resolution is low;

(3) the line number and page number is missing;

(4) Figure2 is so blurry and hard to see;

Reviewer 4 Report

Comments and Suggestions for Authors

This present work reviewed the effect of dislocation slip and twinning on crystal orientation. It is beneficial for deeply and comprehensively understanding the relationship between dislocation slip and twinning and crystal orientation in materials. However, there are lots of textbook style statements in text which is not necessary and academic. The following suggestions should be adopted if this paper could be considered to be published.

1)  Some microstructures of materials which can reveal the crystal orientation should be provided to support the statement of paper.

2)    Some sentences that commonsensical expressions should be deleted or concisely stated. The importance of crystal orientation in materials should be summarized in a separate paragraph.

3)   P2-4, The overview of GNDs is too simple and not deep and comprehensive as a section of paper. Some necessary statements about GNDs in alloys not only in single crystal should be provided.

4)   The quality of some figures must to be improved, for example Figure 2, 6, 17.